# Cryo-EM structure of the CBC-ALYREF complex

**Bradley P Clarke[1], Alexia E Angelos[1], Menghan Mei[1], Pate S Hill[1], Yihu Xie[1]\*, Yi Ren[1,2]\***

[1]Department of Biochemistry, Vanderbilt University School of Medicine Basic Sciences, Nashville, United States; [2]Center for Structural Biology, Vanderbilt University School of Medicine Basic Sciences, Nashville, United States

**Abstract** In eukaryotes, RNAs transcribed by RNA Pol II are modified at the 5′ end with a 7-methylguanosine ($m^7G$) cap, which is recognized by the nuclear cap binding complex (CBC). The CBC plays multiple important roles in mRNA metabolism, including transcription, splicing, polyadenylation, and export. It promotes mRNA export through direct interaction with a key mRNA export factor, ALYREF, which in turn links the TRanscription and EXport (TREX) complex to the 5′ end of mRNA. However, the molecular mechanism for CBC-mediated recruitment of the mRNA export machinery is not well understood. Here, we present the first structure of the CBC in complex with an mRNA export factor, ALYREF. The cryo-EM structure of CBC-ALYREF reveals that the RRM domain of ALYREF makes direct contact with both the NCBP1 and NCBP2 subunits of the CBC. Comparing CBC-ALYREF with other cellular complexes containing CBC and/or ALYREF components provides insights into the coordinated events during mRNA transcription, splicing, and export.

**\*For correspondence:**
yihu.xie@vanderbilt.edu (YX);
yi.ren@vanderbilt.edu (YR)

**Competing interest:** The authors declare that no competing interests exist.

## eLife assessment

This **important** study reports the cryo-electron microscopy structure of a multi-protein complex that recognizes the 5'-end cap of mRNAs and plays a critical role in mRNA export. The structural and biochemical analyses in this study provide **convincing** evidence to support the major claims of the authors, with the inclusion of more functional characterizations in cell-based systems having corroborated the claims further and thus strengthening the study. This paper would be of interest to structural biologists and RNA biologists working on mRNA metabolism.

## Introduction

The nuclear cap binding complex (CBC) binds to the $m^7G$ cap of RNAs transcribed by RNA pol II. It is comprised of NCBP1 (also known as CBP80) and NCBP2 (also known as CBP20). This heterodimeric CBC can form a variety of interactions with different proteins to promote mRNA processing and influence the fate of the transcript. As such, the CBC regulates gene expression at multiple levels, ranging from transcription and splicing to nuclear export and translation (*Gonatopoulos-Pournatzis and Cowling, 2014*; *Rambout and Maquat, 2020*).

The CBC is one of a myriad of protein factors that associate with newly synthesized transcripts. These factors package mRNAs into compacted ribonucleoprotein particles (mRNPs). While the overall structural arrangement of mRNPs is not known, factors such as serine/arginine-rich (SR) proteins are suggested to contribute to mRNP compaction. Prior to export, mRNPs acquire the export receptor NXF1-NXT1 to gain access to the nuclear pore complex (NPC) (*Xie and Ren, 2019*). NXF1-NXT1 can interact with the FG repeats of the nucleoporin proteins in the NPC to mediate mRNA export (*Fribourg et al., 2001*; *Grant et al., 2003*). Intriguingly, early electron microscopic studies on the

Balbiani ring mRNPs in *Chironomus tentans* showed that mRNPs translocate through the NPC in a 5′ to 3′ direction (*Mehlin et al., 1992*). Recent single-molecule work on human mRNPs also suggests mRNA is exported 5′ end first (*Ashkenazy-Titelman et al., 2022*).

The CBC promotes nuclear mRNA export through its interaction with a key mRNA export factor ALYREF. ALYREF in turn binds to DDX39B (also known as UAP56) and by extension, the entire TRanscription-EXport (TREX) complex at the 5′ end of the RNA (*Cheng et al., 2006*). The TREX complex is conserved from yeast to humans (*Strässer et al., 2002*; *Ren et al., 2017*; *Xie et al., 2021a*; *Pacheco-Fiallos et al., 2023*; *Schuller et al., 2020*). The human TREX complex is composed of THOC1, 2, 3, 5, 6, 7, and the DEAD-box helicase DDX39B (*Masuda et al., 2005*). Interestingly, ALYREF was also identified as the THOC4 subunit of the TREX complex, indicating that the function of ALYREF is tightly integrated with the TREX complex. ALYREF and its yeast ortholog Yra1 contain UBMs (UAP56-binding motifs) that mediate the interaction with DDX39B and Sub2, respectively (*Ren et al., 2017*; *Luo et al., 2001*; *Chang et al., 2013*; *Strässer and Hurt, 2001*). The yeast CBC was also shown to facilitate the recruitment of Yra1 onto nascent RNA (*Sen et al., 2019*). ALYREF and TREX play central roles in mRNA export through direct interactions with various factors including the export receptor NXF1-NXT1 (*Viphakone et al., 2012*; *Strässer and Hurt, 2000*; *Stutz et al., 2000*). Their association with the 5′ cap of mRNAs in particular is a key step in the export process. However, thus far the molecular mechanism underlying how ALYREF bridges the CBC and TREX remains unclear.

ALYREF and the associated TREX complex are not only required for cellular mRNA export but also can be hijacked for nuclear export of some viral mRNAs, such as those of herpes viruses. Two well-studied examples are Herpes simplex virus (HSV-1) ICP27 and Herpesvirus saimiri (HVS) ORF57 (*Boyne et al., 2008*; *Sandri-Goldin, 2008*). Both HSV-1 ICP27 and HVS ORF57 directly target the host ALYREF protein (*Tunnicliffe et al., 2011*; *Tunnicliffe et al., 2014*). Structural studies show that they recognize overlapping surfaces on the RRM domain of ALYREF. How these viral factors affect host CBC-ALYREF interaction and function is not known.

The highly integrated nuclear mRNA processing and mRNP packaging also require the actions of multi-functional splicing factors, such as the SR protein SRSF1 and the exon junction complex (EJC). SRSF1 couples transcription, splicing, and export through direct interactions with the CBC, spliceosome, NXF1-NXT1, and RNA (*Huang et al., 2004*; *Cléry et al., 2013*; *Townsend et al., 2020*). Of note, ALYREF also has binding activities to these factors. Therefore, the functions of ALYREF and SRSF1 in mRNA processing and export are likely interconnected. In higher eukaryotes, the EJC is deposited 20–24 nucleotides upstream of the exon junctions during splicing (*Le Hir et al., 2000*; *Andersen et al., 2006*; *Bono et al., 2006*). The EJC was initially shown to associate with ALYREF through a WxHD motif in the N-terminal unstructured region of ALYREF (*Gromadzka et al., 2016*). More recently, multiple binding interfaces were shown between ALYREF and the EJC (*Pacheco-Fiallos et al., 2023*). How the CBC-ALYREF connection affects the function of EJC-ALYREF remains elusive.

To better understand the multiple functions of ALYREF and the CBC in RNA metabolism, we carried out structural and biochemical studies. We present cryo-EM structures of the CBC and CBC-ALYREF complexes at 3.38 Å and 3.22 Å resolution, respectively. The CBC-ALYREF structure reveals that both the NCBP1 and NCBP2 subunits of the CBC interact with ALYREF. HSV-1 ICP27 and HVS ORF57 target the binding interface between the RRM domain of ALYREF and the CBC. We suggest that these viruses not only hijack host pathways to export their own RNA but could also inhibit host RNA metabolism through their interactions with ALYREF. Structural overlay of CBC-ALYREF and EJC-ALYREF reveals that both the CBC and the EJC bind to the RRM domain of ALYREF in a mutually exclusive manner. This suggests that ALYREF's interaction with the EJC is favored after ALYREF dissociates from the CBC, or as an independent event.

## Results and discussion
### ALYREF directly interacts with the CBC

The recombinant human ALYREF protein was shown to interact with the CBC in RNase-treated nuclear extracts (*Cheng et al., 2006*). Here, we used purified recombinant proteins to further investigate the molecular interactions between ALYREF and the CBC (*Figure 1*). It is well known that RS domain-containing proteins, including ALYREF, exhibit low solubility and are prone to aggregation. The addition of glutamic acid and arginine to the buffer can increase protein solubility and stability (*Golovanov*

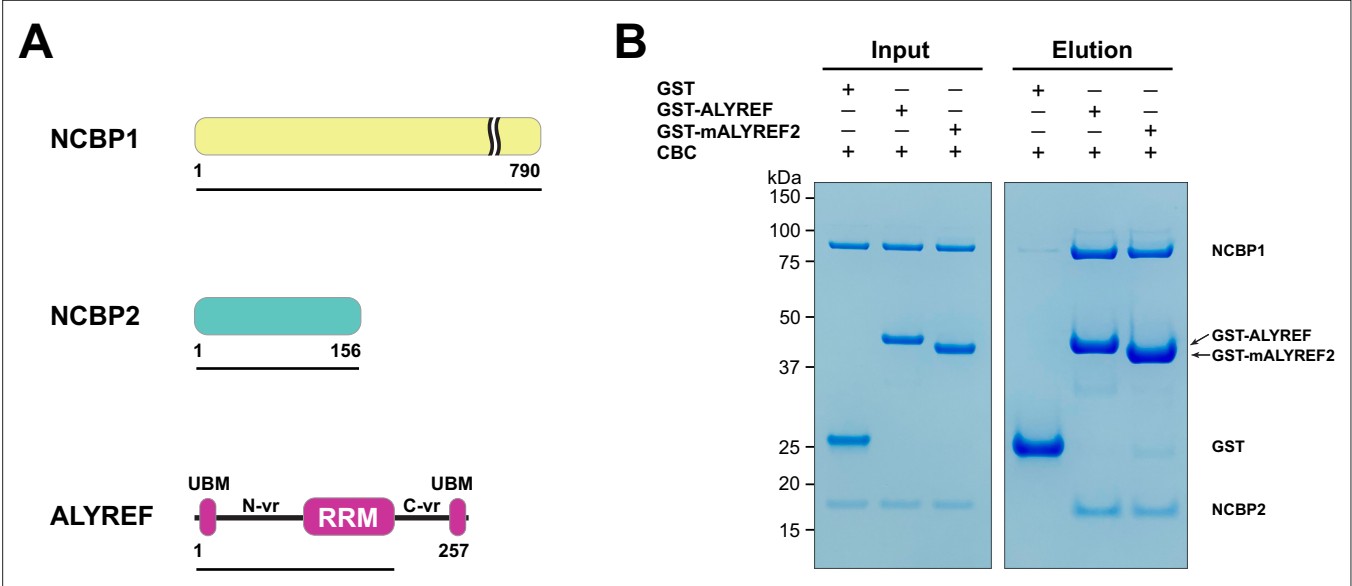

**Figure 1.** ALYREF directly binds to CBC. (**A**) Schematics of the NCBP1 and NCBP2 subunits of the human CBC and ALYREF. ALYREF contains a central RRM domain. The RRM domain is connected to two conserved motifs (UBMs) at both termini through variable regions (N-vr and C-vr). Protein constructs used in GST pull-down assays in panel (**B**) (NCBP1, residues 1–790; NCBP2, residues 1–156; ALYREF, residues 1–183) are indicated by black lines under the respective proteins. (**B**) ALYREF directly interacts with the CBC. In vitro GST pull-down assays were performed with purified recombinant human CBC and GST-tagged ALYREF or the corresponding construct of mouse ALYREF2 (mALYREF2, residues 1–155). Results are representative of three technical repeats.

The online version of this article includes the following source data and figure supplement(s) for figure 1:

**Source data 1.** Original file for the gels in *Figure 1B*.

**Figure supplement 1.** Human ALYREF and mouse ALYREF2 are conserved.

---

*et al., 2004*). Indeed, with a 1:1 mixture of glutamic acid and arginine, we were able to purify a GST-tagged ALYREF (residues 1–183) protein containing the N-terminal region and the RRM domain of ALYREF (*Figure 1A*). This construct includes the WxHD motif (residues 87–90), whose mutation was shown to affect interaction with the CBC in an immunoprecipitation study (*Gromadzka et al., 2016*). GST pull-down assays showed that ALYREF directly interacts with the CBC (*Figure 1B*).

We next attempted to characterize the CBC-ALYREF interaction using cryo-EM. We found that human ALYREF is prone to aggregation in the absence of a GST tag. To obtain an untagged ALYREF protein amenable for structural studies, we tested the solubility of several ALYREF orthologs and found that mouse ALYREF2 (mALYREF2, residues 1–155) exhibited better solubility than the ALYREF construct utilized above. The conserved UBM motifs, the WxHD motif, and the RRM domain are nearly identical between ALYREF and mALYREF2 (*Figure 1—figure supplement 1*). Indeed, like ALYREF, mALYREF2 directly interacts with the CBC (*Figure 1B*). As the ALYREF/mALYREF2 interaction with the CBC is conserved and mALYREF2 exhibits better solubility, we focused on mALYREF2 in the cryo-EM investigations.

## Cryo-EM structures of the CBC and CBC-ALYREF complexes

We collected cryo-EM data from the mixture of the CBC complex and mALYREF2 in the presence of the cap analog m⁷GpppG. The data yielded two maps that correspond to the CBC complex and the CBC-mALYREF2 complex, respectively. The CBC map was refined to an overall resolution of 3.38 Å (*Figure 2A*, *Figure 2—figure supplement 1–3*, *Table 1*). The electron density of the cap analog is clearly visible bound to NCBP2 (*Figure 2—figure supplement 3C*). Compared to unliganded CBC, the cap analog induces significant rearrangements in both the N-terminal extension and the C-terminal tail of NCBP2 (*Figure 2B*) to form critical interactions. For example, the N-terminal extension (residues 16–28) swings toward the central globular domain of NCBP2 and positions the Y20 residue to sandwich the cap analog with Y43 of NCBP2. These conformational changes were also observed in the crystal structures of ligand-bound CBC (*Calero et al., 2002*; *Mazza et al., 2002*). Overall, the



**Figure 2.** Cryo-EM structures of CBC and CBC-mALYREF2. (**A**) Overall architecture of the CBC complex. NCBP1 and NCBP2 are colored in yellow and teal, respectively. The cap analog is shown as orange sticks. (**B**) Comparison of the CBC cryo-EM structure to the unliganded CBC crystal structure (PDB ID 1N54). The cryo-EM structure is colored as in (**A**). The unliganded CBC crystal structure is colored in light blue. Arrow indicates the conformational change in the N-terminal extension of NCBP2 upon cap analog binding. (**C**) Overall architecture of the CBC-mALYREF2 complex. NCBP1, NCBP2, and mALYREF2 are colored in yellow, teal, and red, respectively. The RRM domain of mALYREF2 binds to the CBC. (**D**) Comparison of the CBC-mALYREF2 and the CBC cryo-EM structures. CBC structure is colored in gray.

The online version of this article includes the following figure supplement(s) for figure 2:

**Figure supplement 1.** Workflow for cryo-EM data processing.

**Figure supplement 2.** Cryo-EM reconstruction of the CBC.

**Figure supplement 3.** Structural model of the CBC.

**Figure supplement 4.** Cryo-EM reconstruction of the CBC-mALYREF2 complex.

**Figure supplement 5.** Structural model of the CBC-mALYREF2 complex.

**Table 1.** Cryo-EM data collection, refinement, and validation statistics.

| | CBC-mALYREF2 (EMDB EMD-40739) (PDB 8SRR) | CBC (EMDB EMD-40780) (PDB 8SUY) |
|---|---|---|
| **Data collection and processing** | | |
| Microscope/camera | Glacios/Falcon 4i | |
| Voltage (kV) | 200 | |
| Electron exposure (e–/Å²) | 52 | |
| Defocus range (μm) | –1.0 to –2.0 | |
| Pixel size (Å) | 0.732 | |
| Box size (pixels) | 288 | |
| Initial particle images (no.) | 1,625,826 | |
| Final particle images (no.) | 241,915 | 78,039 |
| Map resolution (masked, Å) | 3.22 | 3.38 |
| Fourier shell correlation (FSC) threshold | 0.143 | 0.143 |
| **Refinement** | | |
| Model resolution (masked, Å) | 3.5 | 3.6 |
| FSC threshold | 0.5 | 0.5 |
| Model composition | | |
| Protein residues | 975 | 895 |
| Ligands | 1 | 1 |
| B factors (Å2) | | |
| Protein | 162.9 | 180.1 |
| Ligand | 157.5 | 163.2 |
| r.m.s. deviations | | |
| Bond lengths (Å) | 0.003 | 0.003 |
| Bond angles (°) | 0.417 | 0.413 |
| Validation | | |
| MolProbity score | 1.34 | 1.48 |
| Clashscore | 6.24 | 7.37 |
| Poor rotamers (%) | 0.7 | 0.9 |
| Ramachandran plot | | |
| Favored (%) | 98.0 | 97.6 |
| Allowed (%) | 2.0 | 2.4 |
| Disallowed (%) | 0.0 | 0.0 |

cryo-EM structure of the CBC determined here resembles the previously reported crystal structures of the liganded CBC, with root mean squared deviation (RMSD) of 0.81 Å and 0.71 Å for NCBP1 and NCBP2, respectively (*Calero et al., 2002*).

The CBC-mALYREF2 map was refined to an overall resolution of 3.22 Å (*Figure 2C*, *Figure 2— figure supplements 1, 4* and *Figure 2—figure supplement 5*, *Table 1*). The cap analog is bound to NCBP2 within the CBC-mALYREF2 complex (*Figure 2—figure supplement 5C*). The structure shows that the RRM domain of mALYREF2 binds to both NCBP1 and NCBP2 subunits (*Figure 2C and D*). The N-terminal region of mALYREF2 (residues 1–73) does not show traceable density and is possibly

disordered. mALRYEF binding does not induce a significant overall conformational change in CBC (*Figure 2D*, *Figure 2—figure supplement 5D*). Comparing CBC and CBC-mALYREF2, NCBP1 and NCBP2 have an RMSD of 0.32 Å and 0.30 Å, respectively. Locally, an NCBP1 loop in proximity to mALYREF2, formed by residues 38–45, becomes more ordered.

## CBC-ALYREF interfaces

The interfaces between mALYREF2 and the CBC involve 16, 18, and 6 residues of mALYREF2, NCBP1, and NCBP2, respectively (*Figure 3*). The RRM domain of mALYREF2 assumes a canonical β1α1β2β3α2β4 topology (*Figure 3A*), forming an α-helical surface and a β-sheet surface. The α-helical surface recognizes the CBC through extensive hydrophilic and hydrophobic interactions (*Figure 3B*, *Figure 3—figure supplement 1A and B*). The α1 helix of mALYREF2 is enriched with acidic residues and makes key hydrophilic interactions with NCBP1. For example, E97 forms salt bridges with K330 and K381 of NCBP1. Y135 on the α2 helix of mALYREF2 makes a hydrogen bond with K330 of NCBP1. In addition, the loop between α2 and β4 of mALYREF2 forms hydrophobic interactions with NCBP1. V138, P139, and L140 of mALYREF2 bind to a hydrophobic pocket on NCBP1 formed by A334, V337, and L382.

The interface between mALYREF2 and NCBP2 is near the $m^7G$ binding pocket (*Figure 3C*, *Figure 3—figure supplement 1C*). The α2 helix of mALYREF2 contacts S13 and Y14 in the N-terminal extension of NCBP2. S13 and Y14 also directly interact with NCBP1 and are thought to enable the hinged motion of the N-terminal extension (residues 16–28) upon binding to the cap (*Mazza et al., 2002*). In addition, the α1 helix of mALYREF2 is in proximity to the R105 and I110 residues of NCBP2. NCBP2 exhibits a positively charged groove extending from the cap binding site, which is suggested to be an RNA binding site (*Calero et al., 2002*). Upon mALYREF2 binding, this groove is buried. Interestingly, mALYREF2 features a positively charged surface near the $m^7G$ site (*Figure 3D*). Conceivably, this positively charged surface on mALYREF2 could serve as an RNA binding site for the nucleotides following the cap.

Based on the CBC-mALYREF2 structure, we generated mutations (mut-1 and mut-2) on the RRM domain of human ALYREF (ALYREF-RRM, residues 103–183) to validate its interaction with the CBC. For ALYREF-RRM-mut-1 (Y166R/V169R/P170R), mutated residues are localized on the α2-β4 loop and correspond to residues Y135/V138/P139 in mALYREF2 (*Figure 3A and B*). For ALYREF-RRM-mut-2 (E124R/E128R), mutated residues are localized on the α1 helix and correspond to residues E93/E97 in mALYREF2 (*Figure 3A and B*). In agreement with the CBC-ALYREF structure, we found that the RRM domain of ALYREF directly interacts with the CBC, albeit with weaker interaction compared to ALYREF (residues 1–183) (*Figures 1B and 4A*). The difference likely results from the WxHD motif (residues 87–90) localized in the N-terminal region of ALYREF. Evidence suggests that mutation of the WxHD motif reduces ALYREF's interaction with the CBC (*Gromadzka et al., 2016*). The WxHD motif may represent a second binding site for the CBC that remains to be characterized. Importantly, compared to the wild type protein, both ALYREF-mut-1 and mut-2 show reduced binding to the CBC (*Figure 4A*). Together, the mutagenesis studies validate the CBC-ALYREF interfaces observed in the structure.

The CBC-mALYREF2 structure reveals that the interaction between ALYREF and the CBC mainly involves the NCBP1 subunit (*Figure 3B and C*). We further dissected the interaction between ALYREF and individual NCBP1 and NCBP2 subunits using GST pull-down assays. NCBP1 can be efficiently pulled down by GST-ALYREF, whereas NCBP2 did not show detectable interaction (*Figure 4B*). These results are consistent with the structural observations and indicate that NCBP1 is the major subunit of the CBC to interact with ALYREF.

## CBC-ALYREF and 5′ cap-dependent mRNP export

ALYREF recruits the mRNP export machinery TREX complex to the 5′ end of mRNA through direct interactions with both the CBC and TREX (*Cheng et al., 2006*; *Ren et al., 2017*). The UBMs of ALYREF directly interact with the DDX39B component of the TREX complex (*Ren et al., 2017*). The N-terminal UBM is included in the ALYREF construct used for our cryo-EM studies but did not show visible electron density. Thus, this UBM is likely exposed and available to interact with DDX39B, which further connects to the entire TREX complex (*Figure 5A*). Consistently, ALYREF, DDX39B, THOC1, and THOC2 are present in NCBP1 immunoprecipitations from RNase-treated HeLa cell nuclear extracts (*Cheng*

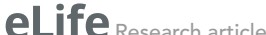

**Figure 3.** ALYREF binds to both NCBP1 and NCBP2. (**A**) Sequence alignment of the RRM domain of mALYREF2 and ALYREF (*Robert and Gouet, 2014*). The residues that interface with the CBC are indicated by triangles below the sequence. The triangles colored in purple correspond to residues subjected to mutagenesis in *Figure 4*. (**B**) Details of the interaction between NCBP1 and mALYREF2. As a reference, an overall model is shown on the top to indicate the zoomed-in area. (**C**) Details of the interaction between NCBP2 and mALYREF2. (**D**) mALYREF2 features a positively charged surface

*Figure 3 continued on next page*

*Figure 3 continued*

near the cap analog bound to NCBP2. The surface of mALYREF2 is colored according to the electrostatic potential, ranging from red (–5 $k_BT/e$) to blue (+5 $k_BT/e$). Dotted line indicates a putative RNA binding path.

The online version of this article includes the following figure supplement(s) for figure 3:

**Figure supplement 1.** Electron density maps at the CBC-mALYREF2 interfaces.

*et al., 2006*). In yeast, mutually exclusive interactions were shown between Yra1 with Sub2 and the NXF1-NXT1 ortholog Mex67-Mtr2 (*Strässer et al., 2002*). So, the ALYREF-dependent NXF1-NXT1 loading on mRNA likely occurs after DDX39B dissociates from ALYREF. The CBC could also function as a landing pad for ALYREF as previously proposed (*Viphakone et al., 2019*). After recruitment to the 5′ end of mRNA by the CBC, ALYREF could then transfer away from the 5′ end, to other sites enriched with export factors and participate in different complexes located along the mRNA. In addition to the ALYREF-NXF1-NXT1 complex, some other ALYREF containing complexes could exist on the same mRNP, such as the complex of ALYREF/DDX39B/SARNP, which facilitates high-order mRNP assembly (*Dufu et al., 2010*; *Xie et al., 2023*).

The process of mRNP export, with the 5′ end exiting first from the NPC, has been shown in both insect and human systems using electron microscopy and single-molecule imaging techniques (*Mehlin et al., 1992*; *Ashkenazy-Titelman et al., 2022*). Interestingly, in the latter study, several adjacent NPCs were found to engage in the export of the same mRNA (*Blobel, 1985*). This observation is reminiscent of the gene gating hypothesis, which suggested that transcriptionally active genes are physically tethered to the site of mRNA export at the NPC (*Blobel, 1985*). Gated genes have been shown in yeast, worms, flies, and humans (*Burns and Wente, 2014*; *Scholz et al., 2019*). For these gated genes, the 5′ directionality of mRNA export could be primarily driven by the key placement of crucial RNA export factors at the 5′ end of the gene as illustrated here (*Figure 5A*), and this localization of export factors could greatly increase the efficiency of co-transcriptional processing and export.

## CBC-ALYREF and viral hijacking of host mRNA export pathway

HSV-1 ICP27 and HVS ORF57 hijack the host mRNA export pathway through interactions with ALYREF. The RRM domain of ALYREF is targeted by both HSV-1 ICP27 and HVS ORF57 with overlapping interfaces (*Figure 5B*, *Figure 5—figure supplement 1*). Structural comparison between CBC-ALYREF, ALYREF-ICP27, and ALYREF-ORF57 reveals that the interface between ALYREF's RRM domain with the CBC is not compatible with the ICP27/ORF57-ALYREF interactions (*Figure 5B*, *Figure 5—figure supplement 1*). In addition, in vivo data show that the ORF57 ortholog from Kaposi's Sarcoma-Associated Herpesvirus (KSHV) can still form a complex with ALYREF and the CBC (*Boyne et al., 2008*). So, it is likely that although ALYREF's RRM domain interface with the CBC could be disrupted by ORF57, ALYREF can still use the WxHD motif to interact with the CBC (*Figure 5C*). Using this strategy, a virus can hijack the host pathway while simultaneously disrupting host interactions and processes. It should also be noted that the CBC, ALYREF, ICP27, and ORF57 are all RNA binding proteins. In addition to the protein-mediated interactions discussed above, RNA interactions should be considered, especially under the in vivo setting (*Figure 5C*). NXF1-NXT1 and DDX39B, the cellular ALYREF interacting proteins, are also hijacked by other factors from viruses. NXF1-NXT1 is targeted by influenza A virus NS1 protein (*Zhang et al., 2019b*) and SARS-CoV-2 Nsp1 protein (*Zhang et al., 2021*; *Mei et al., 2024*). DDX39B is targeted by influenza A virus NP protein (*Momose et al., 2001*; *Morris et al., 2020*). The molecular mechanisms revealed here and from previous studies pave the way for new useful targets in antiviral therapeutics.

## Functional interplay of CBC-ALYREF and mRNP export factors

Transcription, splicing, and export are all tightly linked processes. The CBC promotes splicing through its binding partners, such as SRSF1 (*Lenasi et al., 2011*; *Pabis et al., 2013*). The detailed molecular interaction between the CBC and SRSF1 is revealed in a human pre-B$^{act-1}$ spliceosome structure (*Townsend et al., 2020*). The NCBP2 subunit of the CBC is the major binding site for SRSF1. Structural overlay of this structure with the CBC-ALYREF structure shows no significant steric hindrance between SRSF1 and ALYREF (*Figure 6A*). However, whether the CBC-ALYREF-SRSF1 complex exits in vivo and how their functions might be coordinated require further studies. After splicing, SRSF1 is mainly

**Figure 4.** Dissection of the ALYREF and CBC interfaces. (**A**) Mutations of key interface residues on ALYREF reduced CBC binding. In vitro GST pull-down assays were performed with purified recombinant human CBC and GST-tagged ALYREF-RRM wild type or mutants (mut-1, Y166R/V169R/P170R; mut-2, E124R/E128R). (**B**) NCBP1 is sufficient to interact with ALYREF. In vitro GST pull-down assays were performed with purified recombinant GST-tagged ALYREF and individual CBC subunits. Results are representative of three technical repeats.

The online version of this article includes the following source data for figure 4:

**Source data 1.** Original file for the gels in *Figure 4A*.

**Source data 2.** Original file for the gels in *Figure 4B*.

deposited on exons (*Das and Krainer, 2014*; *Pandit et al., 2013*). SRSF1 functions in mRNA export through interaction with the export receptor NXF1-NXT1 (*Huang et al., 2004*; *Reed and Cheng, 2005*; *Müller-McNicoll et al., 2016*). Of note, the interaction between both ALYREF and SR proteins with NXF1-NXT1 is regulated by phosphorylation. Only hypophosphorylated SR proteins can bind to NXF1-NXT1 efficiently (*Huang et al., 2004*; *Okada et al., 2008*; *Xie et al., 2021b*). Interestingly, SR

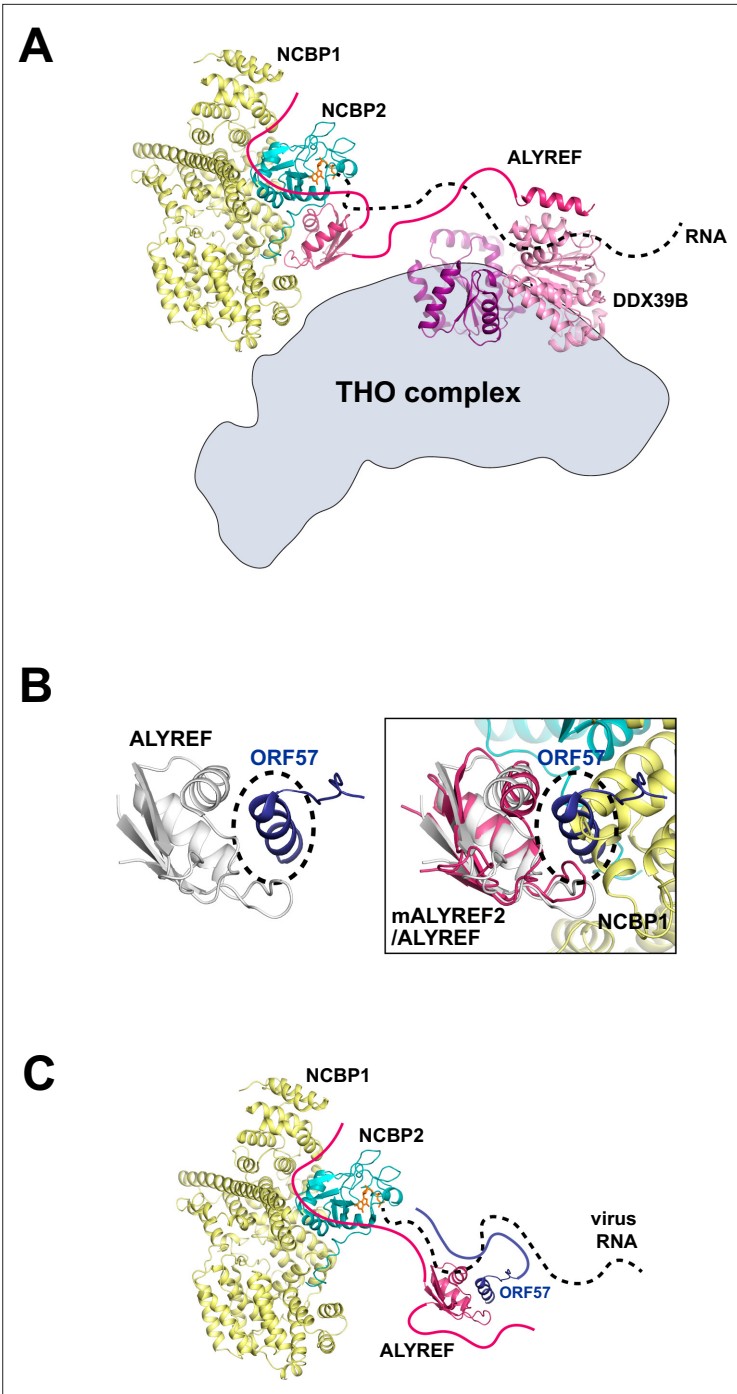

**Figure 5.** Functional implications for ALYREF and CBC in 5' cap-dependent mRNP export. (**A**) ALYREF links the TREX complex to the CBC. The RRM domain of ALYREF recognizes the CBC at the 5' end of mRNA. The UBM of ALYREF binds to the DDX39B subunit of TREX; their complex is represented using their yeast orthologs Yra1 and Sub2, respectively (PDB ID 5SUP). DDX39B in turn associates with the THO subcomplex of TREX. (**B**) HVS ORF57 binds to the RRM domain of ALYREF and interferes with the CBC-ALYREF interaction. (Left) NMR structure of the ALYREF-ORF57 complex (PDB ID 2YKA). (Right) The CBC-ALYREF structure is overlayed with the ALYREF-ORF57 structure. (**C**) Proposed model of viral mRNA export mediated by herpes viral ORF57 homologs. On viral transcripts, ALYREF associates with the CBC and ORF57 via the WxHD motif and the RRM domain, respectively. Both ALYREF and ORF57 feature RNA binding regions to form contacts with the RNA. ALYREF recruits the other TREX complex components to facilitate the nuclear export of viral mRNAs.

The online version of this article includes the following figure supplement(s) for figure 5:

*Figure 5 continued on next page*

**Figure supplement 1.** HSV-1 ICP27 binds to the RRM domain of ALYREF and interferes with the CBC-ALYREF interaction.

**Figure 6.** ALYREF and CBC in splicing and export. (**A**) Overlay of the CBC-ALYREF structure with the CBC-SRSF1 structure (PDB ID 7ABG). CBC-ALYREF is colored as in *Figure 2B*. CBC-SRSF1 is colored with the CBC in white and SRSF1 in orange. (**B**) ALYREF interaction with the CBC and the EJC is mutually exclusive. (Left) ALYREF binds to the MAGOH subunit of the EJC (PDB ID 7ZNJ). (Right) The CBC-ALYREF structure is overlayed with the ALYREF-MAGOH structure. (**C**) Proposed model of the mRNP export receptor NXF1-NXT1 recruitment by CBC-ALYREF and other factors during mRNA maturation.

proteins, such as Gbp2 and Hrb1 in *Saccharomyces cerevisiae*, also interact directly with the TREX complex (*Xie et al., 2021a*; *Hurt et al., 2004*), suggesting coordinated actions of TREX and SR proteins in mRNA export.

Unlike the CBC and SRSF1 interfaces discussed above, the interaction between the ALYREF RRM domain and the CBC is incompatible with the interface between the ALYREF RRM domain and the EJC subunit MAGOH (*Pacheco-Fiallos et al., 2023*; *Figure 6B*). Mutation of the ALYREF WxHD motif affects its interaction with both the CBC and the EJC subunit eIF4A3 (*Gromadzka et al., 2016*). As both the WxHD motif and the RRM domain of ALYREF are mutually exclusive binding sites for the CBC and the EJC, the formation of the EJC-ALYREF complex likely happens after ALYREF dissociates from the CBC. It is also possible that the EJC-ALYREF interaction is independent of the CBC (*Figure 6C*). This possibility is supported by the report that ALYREF can be recruited to RNA by both CBC-dependent and -independent mechanisms (*Nojima et al., 2007*). The resulting mRNP with multiple copies of ALYREF and NXF1-NXT1, each recruited through different mechanisms and at different sites on the mRNP (*Figure 6C*), could exhibit increased export efficiency.

## Conclusion and perspectives

The CBC plays important roles in multiple steps of mRNA metabolism through interactions with a plethora of factors. Here, we present the structural basis of the interaction between the CBC and a key mRNA export factor, ALYREF. The CBC-ALYREF structure reveals molecular insights into the ALYREF-mediated recruitment of mRNA export machinery to the 5′ end of nascent transcripts. We suggest working models for the coordinated events mediated by the CBC and ALYREF during splicing and mRNA export. Notably, both the CBC and ALYREF have been implicated in cancer. Mutations of the CBC residues interfacing with ALYREF, including K330N of NCBP1, R105C of NCBP2, and I110M of NCBP2, are found in several forms of cancers (*Tate et al., 2019*; *Heath et al., 2021*; *Zhang et al., 2019a*). These mutations could reduce ALYREF interaction and subsequently cause dysregulation of mRNA export and the processes that are coordinated by CBC and ALYREF. In addition, ALYREF has been shown to be frequently upregulated in various cancerous tissues (*Domínguez-Sánchez et al., 2011*). Determining whether dysregulation of the CBC-ALYREF interaction contributes to cancer pathogenesis, including characterization of the impact of cancer-associated mutations of the CBC, is an interesting area for future studies.

To date, it is unclear whether the functional connection between the CBC and ALYREF is conserved in yeast. In humans, recruitment of the mRNA export machinery is mediated by ALYREF in a CBC and EJC-dependent manner (*Cheng et al., 2006*; *Gromadzka et al., 2016*). Our work and studies by others (*Pacheco-Fiallos et al., 2023*; *Gromadzka et al., 2016*) reveal that both the CBC and the EJC recognize ALYREF through the WxHD motif and the RRM domain of ALYREF. There are notable differences between yeast and humans regarding ALYREF function: (1) the EJC is not present in the budding yeast, *S. cerevisiae*, and (2) the yeast ortholog of ALYREF, Yra1, does not contain a WxHD motif. Nevertheless, a study conducting transcriptome-wide mapping of mRNA biogenesis factors showed that Yra1 is enriched at the 5′ region of the mRNA (*Baejen et al., 2014*), raising the possibility that Yra1 localization could be mediated by the CBC. Further studies are required to reveal the mechanism of Yra1 in the context of the CBC and recruitment of the mRNA export machinery in yeast.

## Materials and methods

### Plasmids

Human NCBP1 (UniProt Q09161) was cloned into the pFastBac HTc vector with an N-terminal TEV cleavable His tag. Human NCBP2 (UniProt P52298) was cloned into a modified pFastBac1 vector containing an N-terminal TEV cleavable GST tag. Human ALYREF (UniProt Q86V81) and mouse ALYREF2 (UniProt Q4KL64) constructs were cloned into a modified pGST-4T-1 vector containing an N-terminal TEV cleavable GST tag.

### Protein expression and purification

The NCBP1-NCBP2 complex was expressed in High-Five insect cells (Invitrogen) by coinfection of recombinant baculoviruses. Individual NCBP1 and NCBP2 subunits were expressed in High-Five insect cells infected with the respective recombinant baculovirus. High-Five cells were harvested 48 hr after

infection and lysed in a buffer containing 50 mM Tris pH 8.0, 300 mM NaCl, 0.2 mM AEBSF, 2 mg/L aprotinin, 1 mg/L pepstatin, 1 mg/L leupeptin, and 0.5 mM TCEP. Proteins were purified using Glutathione Sepharose 4B resin (Cytiva) for NCBP2 and NCBP1-NCBP2, or Ni Sepharose 6FF resin (Cytiva) for NCBP1 alone. The proteins were further purified on a mono Q column (Cytiva). The expression tags were removed by overnight incubation with GST-TEV (for NCBP2 and NCBP1-NCBP2) or His-TEV (for NCBP1) at 4°C. The samples were passed through Glutathione Sepharose 4B resin (for NCBP2 and NCBP1-NCBP2) or Ni Sepharose 6FF resin (for NCBP1) to remove undigested protein and TEV. The proteins were further purified on a Superdex 200 column (Cytiva) in 10 mM Tris pH 8.0, 300 mM NaCl, and 0.5 mM TCEP.

GST-tagged ALYREF (residues 1–183), ALYREF-RRM (residues 103–183), ALYREF-RRM-mut-1 (Y166R/V169R/P170R), ALYREF-RRM-mut-2 (E124R/E128R), and mALYREF2 (residues 1–155) were expressed in *Escherichia coli* Rosetta cells (Sigma-Aldrich). Protein expression was induced with 0.5 mM IPTG at 16°C overnight. Cells were lysed in a buffer containing 50 mM Tris, pH 8.0, 500 mM NaCl, 50 mM glutamic acid, 50 mM arginine, 0.5 mM TCEP, 0.2 mM AEBSF, and 2 mg/L aprotinin. The GST-tagged proteins were pulled down using Glutathione Sepharose 4B resin. GST-tagged ALYREF-RRM wild type and mutant proteins were purified on a Superdex 200 column in 10 mM Tris pH 8.0, 300 mM NaCl, and 5 mM DTT. GST-tagged ALYREF (residues 1–183) was purified by a cation exchange column (source 15S, Cytiva), followed by a Superdex 200 column equilibrated with 10 mM Tris, pH 8.0, 500 mM NaCl, 50 mM glutamic acid, 50 mM arginine, and 0.5 mM TCEP. GST-tagged mALYREF2 (residues 1–155) was purified by an anion exchange column (Q Sepharose, Cytiva), followed by a cation exchange column (SP Sepharose, Cytiva). For untagged mALYREF2 used in cryo-EM studies, the GST tag was removed by overnight incubation with GST-TEV at 4°C. Untagged mALYREF2 was further purified on a cation exchange HiTrap SP column (Cytiva), followed by a Superdex 200 column equilibrated with 10 mM Tris, pH 8.0, 150 mM NaCl, 50 mM glutamic acid, 50 mM arginine, and 0.5 mM TCEP.

All purified proteins were flash frozen in liquid nitrogen, and stored at –80 °C.

## Cryo-EM sample preparation and data collection

NCBP1-NCBP2 at 1.2 μM was incubated with mALYREF2 (residues 1–155) at 3.6 μM in the presence of the 5′ cap analog $m^7GpppG$ (NEB) at 500 μM at 4°C for 0.5 hr. The sample was deposited on glow-discharged UltrAuFoil R 1.2/1.3 grids (Quantifoil). Grids were blotted for 6 s with a blotting force of 6 at 4°C and 100% humidity and plunged into liquid ethane using a FEI Vitrobot Mark IV (Thermo Fisher). The data were collected with a Glacios Cryo-TEM (Thermo Fisher) equipped with a Falcon 4i detector (Thermo Fisher). Movies were collected with EPU at a magnification of 190,000×, corresponding to a calibrated pixel size of 0.732 Å/pixel. A total of 5858 movies recorded in EER format were collected with a defocus range from 1.0 μm to 2.0 μm. A full description of the cryo-EM data collection parameters can be found in *Table 1*.

## Cryo-EM image processing and model building

Cryo-EM data were processed with cryoSPARC (*Punjani et al., 2017*). Movies in EER format were gain normalized, aligned, and dose-weighted using patch motion correction, followed by patch CTF estimation. A subset of 100 micrographs was subjected to blob particle picking followed by 2D classification to obtain particles for Topaz training. 1,625,826 particles were picked by Topaz from the entire dataset and were extracted with a box size of 288 × 288 pixels. 1,453,188 particles were retained after 2D classification. The particles were subjected to ab initio model reconstruction and heterogeneous refinement, resulting in one good class from 637,185 particles, which showed robust CBC densities as well as densities corresponding to mALYREF2 along with 'junk' classes. Further heterogeneous refinement resulted in a set of 366,410 particles corresponding to the CBC-mALREF2 complex and a set of 121,056 particles corresponding to the CBC complex. The CBC-mALYREF2 particles were subjected to another round of heterogeneous refinement and homogeneous refinement, which yielded a reconstruction from 241,915 particles with an overall resolution of 3.22 Å. Directional anisotropy of the CBC-mALYREF2 map was assessed using the 3DFSC server (*Tan et al., 2017*; https://3dfsc.salk.edu), which indicates a sphericity of 0.946. The CBC particles were subjected to another round of heterogeneous refinement, homogeneous refinement, and non-uniform refinement, which yielded a reconstruction from 78,039 particles with an overall resolution of 3.38 Å. 3DFSC analysis of the CBC map indicates a sphericity of 0.962.

An initial model of CBC was obtained by docking AlphaFold models of NCBP1 (AF-Q09161-F1) and NCBP2 (AF-P52298-F1). An initial model of CBC-mALYREF2 was obtained by docking AlphaFold models of NCBP1, NCBP2, and mALYREF2 into the cryo-EM density map. The models were adjusted in Coot (*Emsley et al., 2010*), followed by real-space refinement in Phenix (*Afonine et al., 2012*). The final CBC model contains NCBP1, NCBP2, and the m⁷G moiety of m⁷GpppG. The final CBC-mALYREF2 model contains NCBP1, NCBP2, the RRM domain of mALYREF2, and the m⁷G moiety of m⁷GpppG. Figures were prepared using PyMOL (Molecular Graphics System, Schrodinger, LLC) and Chimera (*Luo et al., 2001*).

## GST pull-down

GST or GST-tagged proteins were pre-incubated with GST resin in binding buffer (10 mM Tris pH 8.0, 500 mM NaCl, 50 mM glutamic acid, 50 mM arginine, 0.5 mM TCEP) on ice for 0.5 hr and were mixed with gentle tapping every 3–5 min. The beads were then washed three times with 600 µl of buffer containing 10 mM Tris pH 8.0, 100 mM NaCl (*Figures 1B and 4B*) or 50 mM NaCl (*Figure 4A*), and 0.5 mM TCEP. NCBP1, NCPB2, or the NCBP1-NCBP2 complex was incubated with the 5' cap analog m⁷GpppG at 20 µM, adjusted salt concentration to 100 mM NaCl (*Figures 1B and 4B*) or 50 mM NaCl (*Figure 4A*), and then added to beads. The samples were incubated on ice for 0.5 hr and mixed with gentle tapping every 3–5 min. Beads were then washed three times with 600 µl of wash buffer (10 mM Tris pH 8.0, 50 mM NaCl, 0.5 mM TCEP) before bound proteins were eluted in 10 mM Tris pH 8.0, 500 mM NaCl (*Figures 1B and 4B*) or 150 mM NaCl (*Figure 4A*), 25 mM glutathione, and 0.5 mM TCEP. 6% (*Figures 1B and 4B*) or 3% (*Figure 4A*) of the input and 60% of the eluted proteins were analyzed using Coomassie stained SDS-PAGE gels. The experiments were repeated three times independently.

## Acknowledgements

We thank Melissa Chambers, Scott Collier, and Mariam Haider at the Center for Structural Biology Cryo-EM Facility at Vanderbilt University for assistance in Cryo-EM data collection. We acknowledge the use of the Glacios cryo-TEM, which was acquired by NIH grant S10 OD030292-01. This work was supported by NIH R35 GM133743 (YR) and R01 AI184975 (YR). BPC was in part supported by NIH/NCI training grant T32 CA119925.

## Additional information

### Funding

| Funder | Grant reference number | Author |
|---|---|---|
| National Institutes of Health | R35 GM133743 | Bradley P Clarke<br>Alexia E Angelos<br>Menghan Mei<br>Pate S Hill<br>Yihu Xie<br>Yi Ren |
| National Institutes of Health | T32 CA119925 | Bradley P Clarke |
| National Institutes of Health | R01 AI184975 | Menghan Mei<br>Pate S Hill<br>Yihu Xie<br>Yi Ren |

The funders had no role in study design, data collection and interpretation, or the decision to submit the work for publication.

### Author contributions

Bradley P Clarke, Data curation, Formal analysis, Writing - original draft, Writing - review and editing; Alexia E Angelos, Menghan Mei, Data curation, Formal analysis, Writing - review and editing; Pate S Hill, Data curation; Yihu Xie, Data curation, Formal analysis, Supervision, Writing - original draft,

Writing - review and editing; Yi Ren, Data curation, Formal analysis, Supervision, Funding acquisition, Writing - original draft, Writing - review and editing

### Author ORCIDs
Bradley P Clarke ⬤ https://orcid.org/0000-0002-9413-9905
Pate S Hill ⬤ https://orcid.org/0000-0001-9550-2713
Yi Ren ⬤ https://orcid.org/0000-0003-4531-0910

Reviewer #1 (Public Review): https://doi.org/10.7554/eLife.91432.3.sa1
Reviewer #2 (Public Review): https://doi.org/10.7554/eLife.91432.3.sa2
Reviewer #3 (Public Review): https://doi.org/10.7554/eLife.91432.3.sa3
Author response https://doi.org/10.7554/eLife.91432.3.sa4

## Additional files

### Supplementary files
• MDAR checklist

### Data availability
The coordinates of CBC-ALYREF and CBC have been deposited in PDB under accession codes 8SRR and 8SUY. The corresponding density maps have been deposited in EMDB under accession codes EMD-40739 and EMD-40780. Source data files have been provided for Figures 1 and 4.

The following datasets were generated:

| Author(s) | Year | Dataset title | Dataset URL | Database and Identifier |
| --- | --- | --- | --- | --- |
| Xie Y, Clarke BP, Ren Y | 2024 | Cryo-EM structure of the human cap binding complex (CBC) | https://www.rcsb.org/structure/8SUY | RCSB Protein Data Bank, 8SUY |
| Xie Y, Clarke BP, Ren Y | 2024 | Cryo-EM structure of the CBC-ALYREF complex | https://www.rcsb.org/structure/8SRR | RCSB Protein Data Bank, 8SRR |
| Xie Y, Clarke BP, Ren Y | 2024 | Cryo-EM structure of the CBC-ALYREF complex | https://www.ebi.ac.uk/emdb/EMD-40739 | Electron Microscopy Data Bank, EMD-40739 |
| Xie Y, Clarke BP, Ren Y | 2024 | Cryo-EM structure of the human cap binding complex (CBC) | https://www.ebi.ac.uk/emdb/EMD-40780 | Electron Microscopy Data Bank, EMD-40780 |

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
