## [Editor Report · eLife assessment]

This **important** study reports the cryo-electron microscopy structure of a multi-protein complex that recognizes the 5'-end cap of mRNAs and plays a critical role in mRNA export. The structural and biochemical analyses in this study provide **convincing** evidence to support the major claims of the authors, with the inclusion of more functional characterizations in cell-based systems having corroborated the claims further and thus strengthening the study. This paper would be of interest to structural biologists and RNA biologists working on mRNA metabolism.

---

## [Referee Report · Reviewer #1 (Public Review)]

Summary:

The authors use a combination of biochemistry and cryo-EM studies to explore a complex between the cap binding complex and an RNA binding protein, ALYREF, that coordinates mRNA processing and export.

Strengths:

The biochemistry and structural biology are supported by mutagenesis that tests the model in vitro. The structure provides new insight into how key events in RNA processing and export are likely to be coordinated.

Weaknesses:

The authors provide biochemical studies to confirm the interactions that they identify; however, they do not perform any studies to test these models in cells or explore the consequences for mRNA export from the nucleus. In fact, several of the amino acids that they identified in ALYREF that are critical for the interaction, as determined by their own biochemical studies are conserved in budding yeast Yra1 (residues E124/E128 are E/Q in budding yeast and residues Y135/V138/P139 are F/S/P), where the impact on poly(A) RNA export from the nucleus could be readily evaluated. The authors mention the potential for future studies in the manuscript, but they do not perform any analysis in this study that would explore the contributions of these new interactions.

---

## [Referee Report · Reviewer #2 (Public Review)]

Summary:

In this manuscript, Bradley and his colleagues represented the cryo-EM structure of the nuclear cap-binding complex (CBC) in complex with an mRNA export factor, ALYREF, providing a structural basis for understanding CBC regulating gene expression.

Strengths:

The authors successfully modeled the N-terminal region and the RRM domain of ALYREF (residues 1-183) within the CBC-ALYREF structure, which revealed that both the NCBP1 and NCBP2 subunits of the CBC interact with the RBM domain of ALYREF. Further mutagenesis and pull-down studies provided additional evidence to the observed CBC-ALYREF interface. Additionally, the authors engaged in a comprehensive discussion regarding other cellular complexes containing CBC and/or ALYREF components. They proposed potential models that elucidated coordinated events during mRNA maturation. This study provided structural evidence to show how CBC effectively recruits mRNA export factor machinery, enhancing our understanding of CBC regulating gene expression during mRNA transcription, splicing, and export.

Weaknesses:

Absence of functional data to support the proposed models in this study.

---

## [Referee Report · Reviewer #3 (Public Review)]

Summary:

The authors carried out structural and biochemical studies to investigate the multiple functions of CBC and ALYREF in RNA metabolism.

Strengths:

For the structural study part, the authors successfully revealed how NCBP1 and NCBP2 subunits interact with mALYREF (residues 1-155). Their binding interface was then confirmed by biochemical assays (mutagenesis and pull-down assays) presented in this study.

Weaknesses:

The model derived from the cryo-EM structure will likely need to be validated in future functional studies.

---

## [Author Response]

The following is the authors’ response to the original reviews.

**Public Reviews:**

**Reviewer #1 (Public Reviews):**
Summary:The authors use a combination of biochemistry and cryo-EM studies to explore a complex between the cap-binding complex and an RNA binding protein, ALYREF, that coordinates mRNA processing and export.Strengths:The biochemistry and structural biology are supported by mutagenesis which tests the model in vitro. The structure provides new insight into how key events in RNA processing and export are likely to be coordinated.Weaknesses:The authors provide biochemical studies to confirm the interactions that they identify; however, they do not perform any studies to test these models in cells or explore the consequences of mRNA export from the nucleus. In fact, several of the amino acids that they identified in ALYREF that are critical for the interaction, as determined by their own biochemical studies, are conserved in budding yeast Yra1 (residues E124/E128 are E/Q in budding yeast and residues Y135/V138/P139 are F/S/P), where the impact on poly(A) RNA export from the nucleus could be readily evaluated. The authors could at least mention this point as part of the implications and the need for future studies. No one seems to have yet targeted any of these conserved residues, so this would be a logical extension of the current work.

We thank the reviewer for the feedback on our work. ALYREF coordinates pre-mRNA processing and export through interactions with a plethora of mRNA biogenesis factors including the DDX39B subunit of the TREX complex, CBC, EJC, and 3’ processing factors. ALYREF mediates the recruitment of the TREX complex on nascent transcripts which depends on its interactions with both CBC and EJC. Our work and studies by others indicate that ALYREF uses overlapping interfaces including both the N-terminal WxHD motif and the RRM domain to bind CBC and EJC. Thus, ALYREF mutants deficient in CBC interaction will also disrupt the ALYREF-EJC interaction and are not ideal for functional studies. In addition, the CBC plays important roles in multiple steps of mRNA metabolism through interactions with a plethora of factors, which often interact competitively with CBC. Identification of separation-of-function mutations on CBC or ALYREF that specifically disrupt their interaction but not other cellular complexes containing CBC or ALYREF would be an important future area to test the model in cells.

We appreciate the reviewer’s insightful comments regarding yeast Yra1. Thus far, the physical and functional connection between Yra1 and CBC in yeast has not been demonstrated. There are major differences between yeast Yra1 and human ALYREF. Given the lack of an EJC in *S. cerevisiae*, it is unclear whether Yra1 acts in a similar manner as human ALYREF. In addition, Yra1 does not contain a WxHD motif in its N-terminal unstructured region, which is involved in CBC and EJC interactions in ALYREF. Characterization of the Yra1-CBC interaction will be an interesting future direction. We now include a discussion about yeast Yra1 in the newly added “Conclusion and perspectives” section.

Specific suggestions:The authors could put their work in context by speculating how some of the amino acids that they identify as being critical for the interactions they identify could contribute to cancer. For example, they mention mutations of interacting residues in NCBP2 are associated with human cancers, pointing out that NCBP2 R105C amino acid substitution has been reported in colorectal cancer and the NCBP2 I110M mutation has been found in head and neck cancer. Do the authors speculate that these changes would decrease the interaction between NCBP2 and ALYREF and, if so, how would this contribute to cancer? They also mention that a K330N mutation in NCBP1 in human uterine corpus endometrial carcinoma, where Y135 on the α2 helix of mALYREF2 makes a hydrogen bond with K330 of NCBP1. How do they speculate loss of this interaction would contribute to cancer?

In the revised manuscript, we include a discussion about these CBC mutants found in human cancers in the “Conclusion and perspectives” section. We think some of these CBC mutants, such as NCBP-1 K330N, could reduce interaction with ALYREF. Compromised CBC-ALYREF interaction will affect the recruitment of the TREX complex on nascent transcripts and cause dysregulation of mRNA export. In addition, that could also change the partition of CBC and ALYREF in different cellular complexes and cause perturbation of various steps in mRNA biogenesis that are regulated by CBC and ALYREF. Thus far, it is unclear whether and how loss of the CBC-ALYREF interaction directly contributes to cancer. Our work and that of others provide molecular insights to test in future studies.

**Reviewer #2 (Public Reviews):**
Summary:In this manuscript, Bradley and his colleagues represented the cryo-EM structure of the nuclear cap-binding complex (CBC) in complex with an mRNA export factor, ALYREF, providing a structural basis for understanding CBC regulating gene expression.Strengths:The authors successfully modeled the N-terminal region and the RRM domain of ALYREF (residues 1-183) within the CBC-ALYREF structure, which revealed that both the NCBP1 and NCBP2 subunits of the CBC interact with the RBM domain of ALYREF. Further mutagenesis and pull-down studies provided additional evidence to the observed CBC-ALYREF interface. Additionally, the authors engaged in a comprehensive discussion regarding other cellular complexes containing CBC and/or ALYREF components. They proposed potential models that elucidated coordinated events during mRNA maturation. This study provided good evidence to show how CBC effectively recruits mRNA export factor machinery, enhancing our understanding of CBC regulating gene expression during mRNA transcription, splicing, and export.Weaknesses:No in vivo or in vitro functional data to validate and support the structural observations and the proposed models in this study. Cryo-EM data processing and structural representation need to be strengthened.

We appreciate the reviewer’s comments and suggestions. The fact that ALYREF uses highly overlapped binding interfaces for CBC and EJC interactions prevents us from a clear functional dissection of the ALYREF-CBC interaction using in vitro assays or in cells at the current stage. Please also see our response to Reviewer 1.

In this revised manuscript, we have reprocessed the cryo-EM data using a different strategy which yields significantly improved maps. We have made improvements to the presentation of the structural work based on the reviewer’s specific comments.

**Reviewer #3 (Public Reviews):**
Summary:The authors carried out structural and biochemical studies to investigate the multiple functions of CBC and ALYREF in RNA metabolism.Strengths:For the structural study part, the authors successfully revealed how NCBP1 and NCBP2 subunits interact with mALYREF (residues 1-155). Their binding interface was then confirmed by biochemical assays (mutagenesis and pull-down assays) presented in this study.Weaknesses:The authors did not provide functional data to support their proposed models. The authors should include more details regarding the workflow of their cryo-EM data processing in the figure.

We thank the reviewer for the comments. We completely agree that testing the proposed models in cells would be ideal. However, as we also respond to the other reviewers, functional studies are premature at the current stage because both ALYREF and CBC are components of many cellular complexes that regulate mRNA metabolism. Separation-of-function mutations on CBC or ALYREF first need to be identified in future studies for further investigation. Please also see our response to Reviewer 1.

As suggested by the reviewer, we have included more details of the cryo-EM workflow in this revised manuscript. We have also included various validation measures including 3DFSC analyses, map vs model FSC curves, and representative density maps at various protein-protein binding interfaces.

**Recommendations for the Authors:**

**Reviewer #1 (Recommendations for the Authors):**
Major points:The authors should take advantage of Figure 1, which shows the domain structures of NCBP1, NCBP2, and ALYREF to indicate for the reader specifically which protein domains are included in the biochemical and structural analyses. In the current version of the manuscript, there is plenty of space to indicate below each domain structure precisely what regions are included.

In this revised manuscript, we have revised Figure 1A to indicate the protein constructs used in this work.

Although it is fine to combine the Results and Discussion, the authors should really offer a concluding paragraph to highlight the novel results from this study and put the results in context.

We thank the reviewer for the recommendation. We now include a “Conclusion and perspectives” section in this revised manuscript.

Minor comments:Page 5, last sentence (and others) starts a sentence with the word "Since" when likely "As" which does not imply a time element to the phrase, is the correct word."Since the ALYREF/mALYREF2 interaction with the CBC is conserved and mALYREF2 exhibits better solubility, we focused on mALYREF2 in the cryo-EM investigations."Would be more correct as: "As the ALYREF/mALYREF2 interaction with the CBC is conserved and mALYREF2 exhibits better solubility, we focused on mALYREF2 in the cryo-EM investigations."

We thank the reviewer for the comments. We have made the corrections.

The word 'data' is plural so the sentence at the bottom of p.9 that includes the phrase "...in vivo data shows.." should read "..in vivo data show.."

Corrected in the revised manuscript.

**Reviewer #2 (Recommendations for the Authors):**
Major points:(1) The authors claimed the improved solubility of mouse ALYREF2 (mALYREF2, residues 1-155) compared to the previously employed ALYREF construct. However, human ALYREF has already been purified successfully for pull down assay, indicating soluble human ALYREF obtained, why not use human ALYREF directly? Please clarify.

Pull-down studies were performed with GST-tagged ALYREF. For cryo-EM studies, untagged ALYREF is preferred to avoid potential issues that may arise from the expression tag. However, untagged ALYREF is less soluble than GST-tagged ALYREF and is not amenable for structural studies. We have revised the text to clarify this point.

(2) The authors confirmed CBC-ALYREF interfaces through mutagenesis and pull-down assays in vitro. However, it would be more informative and interesting to include functional assays in vitro or/and in vivo with mutagenesis.

We completely concur with the reviewer that testing the proposed models in vitro and in vivo would be important. However, as we pointed out in our response to public reviews, the highly overlapped binding interfaces on ALYREF for CBC and EJC interactions pose a great challenge for functional studies. Furthermore, both ALYREF and CBC are multifunctional factors and interact with a number of partners. Ideally, separation-of-function mutants that specifically disrupt the CBC-ALYREF interaction but not others need to be identified in future studies in order to perform functional studies.

(3) About cryo-EM data processing and structural representation:

(1) In the description of the cryo-EM data processing, the authors claimed they did heterogeneous refinement, homogenous refinement, and then local refinement. This reviewer is puzzled by this process because the normal procedure should be non-uniform refinement following homogenous refinement. If the authors did not perform non-uniform refinement, they should do it because it would significantly improve the quality and resolution of cryo-EM maps. In addition, the right local refinement should include mask files and only show the density/map of the local region.

We thank the reviewer for the suggestions. In response to the reviewer’s comment on the preferred orientation issue (point 5 below), we reprocessed the cryo-EM data and obtained significantly improved cryo-EM maps. In this revised manuscript, the CBC-mALYREF map was refined using homogeneous refinement; the CBC map was refined using homogenous refinement followed by non-uniform refinement. Refinement masks are included in Figure 2-figure supplement1.

(2) Further local refinements with signal subtraction should be performed to improve the density and resolution of mALYREF2.

We tested local refinement with or without signal subtraction using masks covering mALYREF2 and various regions of CBC. Unfortunately, this approach did not improve the density of mALYREF2. We suspect that the small size of mALYREF2 (77 residues for the RRM domain) and the intrinsic flexibility of CBC are the limiting factors in these attempts.

(3) Figures with cryoEM map showing the side chains of the residues on the CBC-mALYREF2 interface should be included to strengthen the claims. Authors could add the map to Figure 3b/c or present it as a supplementary figure.

We include new supplementary figures (Figure 3-figure supplement 1) to show the electron densities corresponding to the views in Figure 3B and 3C. Residues labeled in Figure 3B and 3C are shown in sticks in these supplementary figures.

(4) For cryo-EM date processing, the authors have omitted lots of important details. Could the authors elaborate on the data processing with more details in the corresponding Figure and Methods Sections? Only one abi-initial model from the picked good particles was displayed in the figure. Are there any other different conformations of 3D classes for the dataset? In addition, too few classes have been considered in 3D classification, more classes may give a class with better density and resolution.

We thank the reviewer for the comments. We have reprocessed the cryo-EM data. A major change is to use Topaz for particle picking. We now include more details for data processing in Figure 2-figure supplement 1 and the method section. The cryo-EM sample is relatively uniform. Ab-initio reconstruction and heterogenous refinement yielded only one good class and the other classes are “junk” classes (omitted in the workflow figure). No major conformational changes were observed throughout the multiple rounds of heterogenous refinement for both CBC and CBCmALYREF2. In this revised manuscript, we have been able to obtain significantly improved maps through the new data processing strategy employing Topaz as illustrated in Figure 2-figure supplement 1 to 5.

(5) Angular distribution plots should be included to show if there is a preferred orientation issue. Based on the presented maps in validation reports, there may exist a preferred orientation issue for the reported two cryo-EM maps. Detailed 3D-Histogram and directional FSC plots for all the cryo-EM maps using 3DFSC web server should be presented to show the overall qualities (https://www.nature.com/articles/nmeth.4347 and https://3dfsc.salk.edu/).

We thank the reviewer for the recommendations. In response to the reviewer’s comment on the preferred orientation issue, we reprocessed the cryo-EM data. Topaz was used for particle picking instead of template picking. 3DFSC analyses indicate that the new CBC-mALREF2 map has a sphericity of 0.946, which is a significant improvement from the previous map which has a sphericity of 0.815. Consistently, the maps presented in this revised manuscript show significantly improved densities. We now include angular distribution and 3DFSC analyses of the EM maps (Figure 2-figure supplement 2 and 4).

(6) Figures of model-to-map FSCs need to be present to demonstrate the quality of the models and the corresponding ones (model resolution when FSC=0.5) should also be included in Table 1. The accuracy of the model is important for structural explanations and description.

The model-to-map FSCs are now included in Figure 2-figure supplement 3A and 5A. The model resolutions of CBC-mALYREF2 and CBC are estimated to be 3.5 Å and 3.6 Å at an FSC of 0.5. These numbers are now included in Table 1.

(7) In addition, figures of local density maps with different regions of the models, showing side chains, are necessary and important to justify the claimed resolutions.

We now include density maps overlayed with residue side chains at various regions. For the CBCmALYREF2 map, density maps are shown at the mALYREF2-NCBP1 interfaces (Figure 3-figure supplement 1A and 1B), mALYREF2-NCBP2 interface (Figure 3-figure supplement 1C), NCBP1NCPB2 interface (Figure 2-figure supplement 5B), and the region near m7G (Figure 2-figure supplement 5C). For the CBC map, density maps are shown at the NCBP1-NCPB2 interface (Figure 2-figure supplement 3B) and the region near m7G (Figure 2-figure supplement 3C).

Minor points:(1) A figure superimposing the models from the CBC-mALYREF2 amp and mALYREF2 alone map is necessary to present that there are no other CBC binding-induced conformational changes in CBC except the claimed by the authors. In addition, a figure showing the density of m7GpppG should be included as well.

Overlay of CBC and CBC-mALYREF2 models is now presented in Figure 2-figure supplement 3D. Comparing CBC and CBC-mALYREF2, NCBP1 and NCBP2 have a RMSD of 0.32 Å and 0.30 Å, respectively. The density maps near the M7G cap analog are shown in Figure 2-figure supplement 3C for CBC and Figure 2-figure supplement 5C for CBC-mALYREF2.

(2) Authors obtained the two maps from one dataset, so "we first determined" and "we next determined" (page 6) should be replaced with something like "One class of 3D cryo-EM map revealed' and "Another class of 3D cryo-EM map defined".

We have revised the text as suggested by the reviewer.

(3) In 'Abstract', 'a mRNA export factor' should be 'an mRNA export factor'.

Corrected in the revised manuscript.

(4) In 'Abstract', the final sentence 'Comparison of CBC- ALYREF to other CBC and ALYREF containing cellular complexes provides insights into the coordinated events during mRNA transcription, splicing, and export' doesn't read smoothly, I would suggest revising it to 'Comparing CBC-ALYREF with other cellular complexes containing CBC and/or ALYREF components provides insight into the coordinated events during mRNA transcription, splicing, and export.'

We thank the reviewer for the recommendation and have revised accordingly.

(5) In paragraph 'CBC-ALYREF and viral hijacking of host mRNA export pathway', line 6, the sentences preceding and following the term 'However' indicate a progressive or parallel relationship, rather than a transitional one. To enhance the coherence, I would suggest replacing 'However' with 'Furthermore' or 'In addition'.

Corrected in the revised manuscript.

(6) In both Figure 5 and Figure 6, the depicted models are proposed and constructed exclusively through the comparison of the CBC-partial ALYREF with other cellular complexes containing components of CBC and/or ALYREF, which need to be confirmed by more studies. To prevent potential confusion and misunderstandings, it is recommended to replace the term 'model' with 'proposed model'.

Corrected in the revised manuscript.

**Reviewer #3 (Recommendations for the Authors):**
Major points:(1) In the Results and Discussion section, the authors mentioned "Recombinant human ALYREF protein was shown to interact with the CBC in RNase-treated nuclear extracts." However, they used mouse ALYREF for cryo-EM investigations. Can the authors include an explanation for this choice during the revision?

In our work, we used a mixture of glutamic acid and arginine to increase the solubility of GSTALYREF. For cryo-EM studies, we use untagged ALYREF to avoid potential issues that may arise from the expression tag. However, untagged ALYREF is less soluble than GST-tagged ALYREF and is not suitable for structural studies in standard buffers. We have made further clarification on this point in this revised manuscript.

(2) In the paragraph on "CBC-ALYREF interfaces", the authors stated "For example, E97 forms salt bridges with K330 and K381 of NCBP1. Y135 on the α2 helix of mALYREF2 makes a hydrogen bond with K330 of NCBP1. The importance of this interface between ALYREF and NCBP1 is highlighted by a K330N mutation found in human uterine corpus endometrial carcinoma." I fail to see a strong connection between their structural observations and previous findings regarding the role of a K330N mutation found in human uterine corpus endometrial carcinoma. The authors should add more words to thread these two parts.

In response to the reviewer’s comment, we now move the discussion of these CBC mutants to the newly added “Conclusion and perspectives” section.

(3) The authors should include side chains of the residues in their figure of Local resolution estimation and FSC curves, especially when they are presenting the binding interface between two components.

We have now included density maps that are overlayed with structural models showing side chains of critical residues. These maps include the NCBP1-mALYREF2 interfaces (Figure 3-figure supplement 1A and 1B), NCBP2-mALYREF2 interface (Figure 3-figure supplement 1C), NCBP1NCBP2 interface (Figure 2-figure supplement 3B and 5B), and the m7G cap region (Figure 2figure supplement 3C and 5C).

Minor points:(1) Some grammatical mistakes need to be corrected. For example, it is "an mRNA" instead of "a mRNA".

Corrected in the revised manuscript.

(2) The authors can provide more information for the audience to know better about ALYREF when it first appears in the 5th line in the Abstract section. For example, "It promotes mRNA export through direct interaction with ALYREF, a key mRNA export factor, ...".

We have revised the sentence based on the reviewer’s comment.